# CONQUR: Mitigating Delusional Bias in Deep Q-learning

## Abstract

*Delusional bias* is a fundamental source of error in approximate Q-learning. To date, the only techniques that explicitly address delusion require comprehensive search using tabular value estimates. In this paper, we develop efficient methods to mitigate delusional bias by training Q-approximators with labels that are "consistent" with the underlying greedy policy class. We introduce a simple penalization scheme that encourages Q-labels used *across training batches* to remain (jointly) consistent with the expressible policy class. We also propose a search framework that allows multiple Q-approximators to be generated and tracked, thus mitigating the effect of premature (implicit) policy commitments. Experimental results demonstrate that these methods can improve the performance of Q-learning in a variety of Atari games, sometimes dramatically.

## 1 Introduction

*Q-learning* (Watkins & Dayan, 1992; Sutton & Barto, 2018) lies at the heart of many of the recent successes of deep reinforcement learning (RL) (Mnih et al., 2015; Silver et al., 2016), with recent advancements (e.g., van Hasselt (2010); Bellemare et al. (2017); Wang et al. (2016); Hessel et al. (2017)) helping to make it among the most widely used methods in applied RL. Despite these successes, many properties of Q-learning are poorly understood, and it is challenging to successfully apply deep Q-learning in practice. When combined with function approximation, Q-learning can become unstable (Baird, 1995; Boyan & Moore, 1995; Tsitsiklis & Roy, 1996; Sutton & Barto, 2018). Various modifications have been proposed to improve convergence or approximation error (Gordon, 1995; 1999; Szepesvári & Smart, 2004; Melo & Ribeiro, 2007; Maei et al., 2010; Munos et al., 2016); but it remains difficult to reliably attain both robustness and scalability.

Recently, Lu et al. (2018) identified a source of error in Q-learning with function approximation known as *delusional bias*. It arises because Q-learning updates the value of state-action pairs using estimates of (sampled) successor-state values that can be *mutually inconsistent given the policy class induced by the approximator*. This can result in unbounded approximation error, divergence, policy cycling, and other undesirable behavior. To handle delusion, the authors propose a *policy-consistent backup* operator that maintains multiple Q-value estimates organized into *information sets*. Each information set has its own backed-up Q-values and corresponding "policy commitments" responsible for inducing these values. Systematic management of these sets ensures that only *consistent* choices of maximizing actions are used to update Q-values. All potential solutions are tracked to prevent premature convergence on any specific policy commitments. Unfortunately, the proposed algorithms use tabular representations of Q-functions, so while this establishes foundations for delusional bias, the function approximator is used neither for generalization nor to manage the size of the state/action space. Consequently, this approach is not scalable to RL problems of practical size.

In this work, we propose CONQUR (*CONsistent Q-Update Regression*), a general framework for integrating policy-consistent backups with regression-based function approximation for Q-learning and for managing the search through the space of possible regressors (i.e., information sets). With suitable search heuristics, our framework provides a computationally effective means for minimizing the effects of delusional bias in Q-learning, while admitting scaling to practical problems.

Our main contributions are as follows. First we define novel augmentations of standard Q-regression to increase the degree of policy consistency across training batches. While testing exact consistency

is expensive, we introduce an efficient *soft-consistency penalty* that promotes consistency of new labels with earlier policy commitments. Second, drawing on the information-set structure of Lu et al. (2018), we define a search space over Q-regressors to allow consideration of multiple sets of policy commitments. Third, we introduce heuristics for guiding the search over regressors, which is critical given the combinatorial nature of information sets. Finally, we provide experimental results on the Atari suite (Bellemare et al., 2013) demonstrating that CONQUR can offer (sometimes dramatic) improvements over Q-learning. We also show that (easy-to-implement) consistency penalization on its own (i.e., without search) can improve over both standard and double Q-learning.

## 2 BACKGROUND

We assume a discounted, infinite horizon *Markov decision process (MDP)*, $\mathbf{M} = (\mathcal{S}, A, P, p_0, R, \gamma)$. The state space $\mathcal{S}$ can reflect both discrete and continuous features, but we take the action space $A$ to be finite (and practically enumerable). We consider *Q-learning* with a function approximator $Q_\theta$ to learn an (approximately) optimal Q-function (Watkins, 1989; Sutton & Barto, 2018), drawn from some approximation class parameterized by $\Theta$ (e.g., the weights of a neural network). When the approximator is a deep network, we generically refer to the algorithm as *DQN*, the method at the heart of many recent RL successes (Mnih et al., 2015; Silver et al., 2016).

For online Q-learning, at a transition $s, a, r, s'$, the Q-update is given by:

$$\theta \leftarrow \theta + \alpha \Big( r + \gamma \max_{a' \in A} Q_\theta(s', a') - Q_\theta(s, a) \Big) \nabla_\theta Q_\theta(s, a). \tag{1}$$

Batch versions of Q-learning, including DQN, are similar, but fit a regressor repeatedly to batches of training examples (Ernst et al., 2005; Riedmiller, 2005). Batch methods are usually more data efficient and stable than online Q-learning. Abstractly, batch Q-learning works through a sequence of (possibly randomized) data batches $D_1, \cdots D_T$ to produce a sequence of regressors $Q_{\theta_1}, \ldots, Q_{\theta_T} = Q_\theta$, estimating the Q-function.[1] For each $(s, a, r, s') \in D_k$, we use a prior estimator $Q_{\theta_{k-1}}$ to bootstrap the *Q-label* $q = r + \gamma \max_{a'} Q_{\theta_{k-1}}(s', a')$. We then fit $Q_{\theta_k}$ to this training data using a suitable regression procedure with an appropriate loss function. Once trained, the (implicit) induced policy $\pi_\theta$ is the *greedy policy* w.r.t. $Q_\theta$, i.e., $\pi_\theta(s) = \arg\max_{a \in A} Q_\theta(s, a)$. Let $\mathcal{F}(\Theta)$, resp. $G(\Theta)$, be the corresponding class of expressible Q-functions, resp. greedy policies.

Intuitively, *delusional bias* occurs whenever a backed-up value estimate is derived from action choices that are not (jointly) realizable in $G(\Theta)$ (Lu et al., 2018). Standard Q-updates back up values for each $(s, a)$ pair by *independently* choosing maximizing actions at the corresponding next states $s'$. However, such updates may be "inconsistent" under approximation: if no policy in $G(\Theta)$ can jointly express all past action choices, backed up values may not be realizable by any expressible policy. Lu et al. (2018) show that delusion can manifest itself with several undesirable consequences. Most critically, it can prevent Q-learning from learning the optimal representable policy in $G(\Theta)$; it can also cause divergence. To address this, they propose a non-delusional *policy consistent* Q-learning (PCQL) algorithm that provably eliminates delusion. We refer to the original paper for details, but review the main concepts we need to consider below.

The first key concept is that of *policy consistency*. For any $S \subseteq \mathcal{S}$, an *action assignment* $\sigma_S : S \to A$ associates an action $\sigma(s)$ with each $s \in S$. We say $\sigma$ is *policy consistent* if there is a greedy policy $\pi \in G(\Theta)$ s.t. $\pi(s) = \sigma(s)$ for all $s \in S$. We sometimes equate a set $SA$ of state-action pairs with an implied assignment $\pi(s) = a$ for all $(s, a) \in SA$. If $SA$ contains multiple pairs with the same state $s$, but different actions $a$, it is a *multi-assignment* (though we loosely use the term "assignment" in both cases when there is no risk of confusion).

In (batch) Q-learning, each successive regressor uses training labels generated by assuming maximizing actions (under the prior regressor) are taken at its successor states. Let $\sigma_k$ reflect the collection of states and corresponding maximizing actions taken to generate labels for regressor $Q_{\theta_k}$ (assume it is policy consistent). Suppose we train $Q_{\theta_k}$ by bootstrapping on $Q_{\theta_{k-1}}$ and consider a training sample $(s, a, r, s')$. Q-learning generates label $r + \gamma \max_{a'} Q_{\theta_{k-1}}(s', a')$ for input $(s, a)$. Notice,

---

[1] We describe our approach using a straightforward form of batch Q-learning, but it can accommodate many variants, e.g., where the regressor used for bootstrapping is some earlier Q-estimator, or the estimators generating the max-actions and the value estimates are different as in double Q-learning (van Hasselt, 2010; Hasselt et al., 2016); indeed, we experiment with such variants.

however, that taking action $a^* = \operatorname{argmax}_{a'} Q_{\theta_k}(s', a')$ at $s'$ may not be *policy consistent* with $\sigma_k$. Thus Q-learning will estimate a value for $(s, a)$ assuming the execution of a policy that cannot be realized given the limitations of the approximator. The PCQL algorithm (Lu et al., 2018) prevents this by insisting that any *action assignment* $\sigma$ used to generate bootstrapped labels is consistent with earlier assignments. Notice that this means Q-labels will often *not* be generated using maximizing actions relative to the prior regressor.

The second key concept is that of *information sets*. One will generally not be able to use maximizing actions to generate labels, so tradeoffs can be made when deciding which actions to assign to different states. Indeed, even if it is feasible to assign a maximizing action $a$ to state $s$ early in training, say at batch $k$, since it may prevent assigning a maximizing $a'$ to $s'$ later, say batch $k + \ell$, we may want to consider a different assignment to $s$ to give more flexibility to maximize at other states later. PCQL doesn't try to anticipate the tradeoffs—rather it maintains *multiple information sets*, each corresponding to a different assignment to the states seen in the training data so far. Each gives rise to a *different Q-function estimate*, resulting in multiple hypotheses. At the end of training, the best hypothesis is the one maximizing expected value w.r.t. an initial state distribution.

PCQL provides strong convergence guarantees, but it is a tabular algorithm: the function approximator *retricts* the policy class, but is not used to generalize Q-values. Furthermore, its theoretical guarantees come at a cost: it uses *exact* policy consistency tests—tractable for linear approximators, but not practical for large problems; and it maintains *all* consistent assignments. As a result, PCQL cannot be used for large RL problems of the type tackled by DQN.

## 3 THE CONQUR FRAMEWORK

We develop the CONQUR framework to provide a practical approach to reducing delusion in Q-learning, specifically addressing the limitations of PCQL identified above. CONQUR consists of three main components: a practical soft-constraint penalty that promotes policy consistency; a search space to structure the search over multiple regressors (information sets, action assignments); and heuristic search schemes (expansion, scoring) to find good Q-regressors.

### 3.1 PRELIMINARIES

We assume a set of training data consisting of quadruples $(s, a, r, s')$, divided into (possibly non-disjoint) *batches* $D_1, \dots D_T$ for training. This perspective is quite general: online RL corresponds to $|D_i| = 1$; off-line batch training (with sufficiently exploratory data) corresponds to a single batch (i.e., $T = 1$); and online or batch methods with replay are realized when the $D_i$ are generated by sampling some data source with replacement.

For any data batch $D$, let $\chi(D) = \{s' : (s, a, r, s') \in D\}$ denote the collection of *successor states* of $D$. An *action assignment* $\sigma_D$ for $D$ is an assignment (or multi-assignment) from $\chi(D)$ to $A$: this dictates which action $\sigma_D(s')$ is considered "maximum" for the purpose of generating a Q-label for pair $(s, a)$; i.e., $(s, a)$ will be assigned training label $r + \gamma Q(s', \sigma(s'))$ rather than $r + \gamma \max_{a' \in A} Q(s', a')$. The set of all such assignments is $\Sigma(D) = A^{\chi(D)}$; note that it grows exponentially with $|D|$.

Given Q-function parameterization $\Theta$, we say $\sigma_D$ is $\Theta$-*consistent (w.r.t. $D$)* if there is some $\theta \in \Theta$ s.t. $\pi_\theta(s') = \sigma(s')$ for all $s' \in \chi(D)$.[2] This is simple policy consistency, but with notation that emphasizes the policy class. Let $\Sigma_\Theta(D)$ denote the set of all $\Theta$-consistent assignments over $D$. The union $\sigma_1 \cup \sigma_2$ of two assignments (over $D_1, D_2$, resp.) is defined in the usual way.

### 3.2 CONSISTENCY PENALIZATION

Enforcing strict $\Theta$-consistency as regressors $\theta_1, \theta_2, \dots, \theta_T$ are generated is computationally challenging. Suppose assignments $\sigma_1, \dots, \sigma_{k-1}$, used to generate labels for $D_1, \dots D_{k-1}$, are jointly $\Theta$-consistent (let $\sigma_{\leq k-1}$ denote their multi-set union). Maintaining $\Theta$-consistency when generating $\theta_k$ imposes two requirements. First, one must generate an assignment $\sigma_k$ over $D_k$ s.t. $\sigma_{\leq k-1} \cup \sigma_k$ is consistent. Even testing assignment consistency can be problematic: for linear approximators this is a

---

[2]We suppress mention of $D$ when clear from context or implied by the assignment under consideration.

linear feasibility program (Lu et al., 2018) whose constraint set grows linearly with $|D_1 \cup \ldots \cup D_k|$. For DNNs, this is a complex, and much more expensive, polynomial program. Second, the regressor $\theta_k$ should itself be consistent with $\sigma_{\leq k-1} \cup \sigma_k$. Again, this imposes a significant constraint on the regression optimization: in the linear case, this becomes a constrained least-squares problem (solvable, e.g., as a quadratic program); while with DNNs, it could be solved, say, using a much more complicated projected SGD. However, the sheer number of constraints makes this impractical.

Rather than enforcing consistency, we propose a simple, computationally tractable scheme that "encourages" it: a penalty term that can be incorporated into the regression itself. Specifically, we add a penalty function to the usual squared loss to encourage updates of the Q-regressors to be consistent with the underlying information set, i.e., the prior action assignments used to generate its labels.

When constructing $\theta_k$, let $D_{\leq k} = \cup\{D_j : j \leq k\}$, and $\sigma \in \Sigma_\Theta(D_{\leq k})$ be the collective (possibly multi-) assignment used to generate labels for all prior regressors (including $\theta_k$ itself). The multiset of pairs $B = \{(s', \sigma(s')) | s' \in \chi(D_{\leq k})\}$, is called a *consistency buffer*. The collective assignment need not be consistent (as we elaborate below), nor does the regressor $\theta_k$ need to be consistent with $\sigma$. Instead, we incorporate the following *soft consistency penalty* when constructing $\theta_k$:

$$C_\theta(s', a) = \sum_{a' \in A} [Q_\theta(s', a') - Q_\theta(s', a)]_+ \qquad C_\theta(B) = \sum_{(s', \sigma(s')) \in B} C_\theta(s', \sigma(s')),$$

where $[x]_+ = \max(0, x)$. This penalizes Q-values of actions at state $s$ that are larger than that of action $\sigma(s)$. We note that $\sigma$ is $\Theta$-consistent *if and only if* $\min_{\theta \in \Theta} C_\theta(B) = 0$. We incorporate this penalty into our regression loss for batch $D_k$:

$$L_\theta(D_k, B) = \sum_{(s,a,r,s') \in D_k} [r + \gamma Q_{\theta_{k-1}}(s', \sigma(s')) - Q_\theta(s, a)]^2 + \lambda C_\theta(B). \qquad (2)$$

Here $Q_{\theta_k}$ is prior estimator on which labels are bootstrapped (other prior regressors may be used). The penalty effectively acts as a "regularizer" on the squared Bellman error, where $\lambda$ controls the degree of penalization, allowing a tradeoff between Bellman error and consistency with the action assignment used to generate labels. It thus promotes consistency without incurring the expense of testing strict consistency. It is a simple matter to replace the classical Q-learning update (1) with one using a consistency penalty:

$$\theta_k \leftarrow \theta_{k-1} + \left( \sum_{(s,a,r,s') \in D_k} \alpha[r + \gamma Q_{\theta_{k-1}}(s', \sigma(s')) - Q_\theta(s, a)] \nabla_\theta Q_\theta(s, a) \right)$$
$$+ \alpha\lambda \nabla_\theta C_\theta(B)\Big|_{\theta=\theta_{k-1}}. \qquad (3)$$

This scheme is quite general. First, it is agnostic as to how the prior action assignments are made, which can be the standard maximizing action at each stage w.r.t. the prior regressor like in DQN, Double DQN (DDQN) (Hasselt et al., 2016), or other variants. It can also be used in conjunction with a search through alternate assignments (see below).

Second, the consistency buffer $B$ may be populated in a variety of ways. Including all max-action choices from all past training batches promotes full consistency in an attempt to minimize delusion. However, this may be too constraining since action choices early in training are generally informed by very inaccurate value estimates. Hence, $B$ may be implemented in other ways to focus only on more recent data (e.g., with a sliding recency window, weight decay, or subsampling); and the degree of recency bias may adapt during training (e.g., becoming more inclusive as training proceeds and the Q-function approaches convergence). Reducing the size of $B$ also has various computational benefits. We discuss other practical means of promoting consistency in Sec. 5.

The proposed consistency penalty resembles the temporal-consistency loss of Pohlen et al. (2018), but our aims are very different. Their temporal consistency notion penalizes changes in a next state's Q-estimate over all actions, whereas we discourage inconsistencies in the greedy policy induced by the Q-estimator, regardless of the actual estimated values.

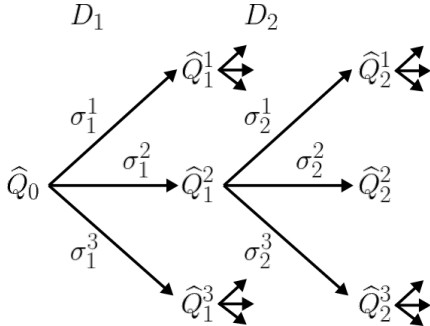

Fig. 1: A generic search tree.

### 3.3 THE SEARCH SPACE

Ensuring optimality requires that PCQL track *all* $\Theta$-*consistent assignments*. While the set of such assignments is shown to be of polynomial size (Lu et al., 2018), it is still impractical to track this set in realistic problems. As such, in CONQUR we recast information set tracking as a *search problem* and propose several strategies for managing the search process. We begin by defining the search space and discussing its properties. We discuss search procedures in Sec. 3.4.

As above, assume training data is divided into batches $D_1, \ldots D_T$ and we have some initial Q-function estimate $\theta_0$ (for bootstrapping $D_1$'s labels). The regressor $\theta_k$ for $D_k$ can, in principle, be trained with labels generated by *any assignment* $\sigma \in \Sigma_\Theta(D_k)$ of actions to its successor states $\chi(D_k)$, not necessarily maximizing actions w.r.t. $\theta_{k-1}$. Each $\sigma$ gives rise to a different updated Q-estimator $\theta_k$. There are several restrictions we could place on "reasonable" $\sigma$-candidates: (i) $\sigma$ is $\Theta$-consistent; (ii) $\sigma$ is jointly $\Theta$-consistent with all $\sigma_j$, for $j < k$, used to construct the prior regressors on which we bootstrap $\theta_{k-1}$; (iii) $\sigma$ is not *dominated* by any $\sigma' \in \Sigma_\Theta(D_k)$, where we say $\sigma'$ dominates $\sigma$ if $Q_{\theta_{k-1}}(s', \sigma'(s')) \geq Q_{\theta_{k-1}}(s', \sigma(s'))$ for all $s' \in \chi(D)$, and this inequality is strict for at least one $s'$. Conditions (i) and (ii) are the strict consistency requirements of PCQL. We will, however, relax these below for reasons discussed in Sec. 3.2. Condition (iii) is inappropriate in general, since we may add additional assignments (e.g., to new data) that render all non-dominated assignments inconsistent, requiring that we revert to some dominated assignment.

This gives us a generic *search space* for finding policy-consistent, delusion-free Q-function, as illustrated in Fig. 1. Each node $n_k^i$ at depth $k$ in the search tree is associated with a regressor $\theta_k^i$ defining $Q_{\theta_k^i}$ and action assignment $\sigma_k^i$ that justifies the labels used to train $\theta_k^i$ ($\sigma_k^i$ can also be viewed as an information set). We assume the root $n_0$ is based on an initial regression $\theta_0$, and has an empty action assignment $\sigma_0$. Nodes at level $k$ of the tree are defined as follows. For each node $n_{k-1}^i$ at level $k-1$—with regressor $\theta_{k-1}^i$ and $\Theta$-consistent assignment $\sigma_{k-1}^i$—we generate a child $n_k^j$ for each $\sigma_k^j \in \Sigma_\Theta(D_k)$ such that $\sigma_{k-1}^i \cup \sigma_k^j$ is $\Theta$-consistent. Node $n_k^j$'s assignment is $\sigma_{k-1}^i \cup \sigma_k^j$, and its regressor $\theta_k^i$ is trained using the following data set:

$$\{(s, a) \mapsto r + \gamma Q_{\theta_{k-1}^i}(s', \sigma_k^j(s')) \ : \ (s, a, r, s') \in D_k\}.$$

The entire search space constructed in this fashion to a maximum depth of $T$. See Appendix B, Algorithm 1 for pseudocode of a simple depth-first recursive specification.

The exponential branching factor in this search tree would appear to make complete search intractable; however, since we only allow $\Theta$-consistent "collective" assignments we can bound the size of the tree—it is *polynomial* in the VC-dimension of the approximator.

**Theorem 1.** *The number of nodes in the search tree is no more than $O(nm \cdot [\binom{m}{2}n]^{\mathsf{VCDim}(\mathcal{G})})$ where* $\mathsf{VCDim}(\cdot)$ *is the VC-dimension (Vapnik, 1998) of a set of boolean-valued functions, and $\mathcal{G}$ is the set of boolean functions defining all feasible greedy policies under $\Theta$:*

$$\mathcal{G} = \{g_\theta(s, a, a') := \mathbf{1}[f_\theta(s, a) - f_\theta(s, a') > 0], \forall s, a \neq a' \mid \theta \in \Theta\}. \tag{4}$$

A linear approximator with a fixed set of $d$ features induces a policy-indicator function class $\mathcal{G}$ with VC-dimension $d$, making the search tree polynomial in the size of the MDP. Similarly, a fixed ReLU

DNN architecture with $W$ weights and $L$ layers has VC-dimension of size $O(WL \log W)$ again rendering the tree polynomially sized.

Even with this bound, navigating the search space exhaustively is generally impractical. Instead, various search methods can be used to explore the space, with the aim of reaching a "high quality" regressor at some leaf of the tree (i.e., trained using all $T$ data sets/batches). We discuss several key considerations in the next subsection.

## 3.4 SEARCH HEURISTICS

Even with the bound in Theorem 1, traversing the search space exhaustively is generally impractical. Moreover, as discussed above, enforcing consistency when generating the children of a node, and their regressors, may be intractable. Instead, various search methods can be used to explore the space, with the aim of reaching a "high quality" regressor at some (depth $T$) leaf of the tree. We outline three primary considerations in the search process: child generation, node evaluation or scoring, and the search procedure.

**Generating children.** Given node $n_{k-1}^i$, there are, in principle, exponentially many action assignments, or children, $\Sigma_\Theta(D_k)$ (though Theorem 1 significantly limits the number of children if we enforce consistency). For this reason, we consider heuristics for generating a small set of children. Three primary factors drive these heuristics.

The first factor is a preference for generating *high-value assignments*. To accurately reflect the intent of (sampled) Bellman backups, we prefer to assign actions to state $s' \in \chi(D_k)$ with larger predicted Q-values over actions with lower values, i.e., a preference for $a$ over $a'$ if $Q_{\theta_{k-1}^j}(s', a) > Q_{\theta_{k-1}^j}(s', a')$. However, since the maximizing assignment may be $\Theta$-inconsistent (in isolation, or jointly with the parent's information set, or with future assignments), candidate children should merely have higher probability of a high-value assignment. The second factor is the need to ensure *diversity* in the assignments among the set of children. Policy commitments at stage $k$ constrain the possible assignments at subsequent stages. In many search procedures (e.g., beam search), we avoid backtracking, so we want the policy commitments we make at stage $k$ to offer as much flexibility as possible in later stages. The third is the degree to which we enforce consistency.

There are several ways to generate such high-value assignments. We focus on just one natural technique: sampling action assignments using a Boltzmann distribution. Specifically, let $\sigma$ denote the assignment (information set) of some node (parent) at level $k-1$ in the tree. We can generate an assignment $\sigma_k$ for $D_k$ as follows. Assume some permutation $s'_1, \ldots, s'_{|D_k|}$ of $\chi(D_k)$. For each $s'_i$ in turn, we sample $a_i$ with probability proportional to $e^{\tau Q_{\theta_{k-1}}(s'_i, a_i)}$. This can be done *without regard to consistency*, in which case we would generally use the consistency penalty when constructing the regressor $\theta_k$ for this child to "encourage" consistency rather than enforce it. If we want strict consistency, we can use rejection sampling without replacement to ensure $a_i$ is consistent with $\sigma_{k-1}^j \cup \sigma_{\leq i-1}$ (we can also use a subset of $\sigma_{k-1}^j$ as a less restrictive consistency buffer).[3] The temperature parameter $\tau$ controls the degree to which we focus on purely maximizing assignments versus more diverse, random assignments. While stochastic sampling ensures some diversity, this procedure will bias selection of high-value actions to states $s' \in \chi(D_k)$ that occur early in the permutation. To ensure sufficient diversity, we use a new random permutation for each child.

**Scoring children.** Once the children of some expanded node are generated (and, optionally, their regressors constructed), we need some way of evaluating the quality of each child as a means of deciding which new nodes are most promising for expansion. Several techniques can be used. We could use the average Q-label (overall, or weighted using some initial state distribution), Bellman error, or loss incurred by the regressor (including the consistency penalty or other regularizer). However, care must be taken when comparing nodes at different depths of the search tree, since deeper nodes have a greater chance to accrue rewards or costs—simple calibration methods can be used. Alternatively, when a simulator is available, rollouts of the induced greedy policy can be used evaluate the quality of a node/regressor. Notice that using rollouts in this fashion incurs considerable computational expense during training relative to more direct scoring based on properties on the node, regressor, or information set.

---

[3]Notice that at least one action for state $s'_i$ must be consistent with any previous (consistent) information set.

**Search Procedure.** Given any particular way of generating/sampling and scoring children, a variety of different search procedures can be applied: best-first search, beam search, local search, etc. all fit very naturally within the CONQUR framework. Moreover, hybrid strategies are possible—one we develop below is a variant of beam search in which we generate multiple children only at certain levels of the tree, then do "deep dives" using consistency-penalized Q-regression at the intervening levels. This reduces the size of the search tree considerably and, when managed properly, adds only a constant-factor (proportional to beam size) slowdown to standard Q-learning methods like DQN.

### 3.5 A CONCRETE INSTANTIATION OF THE CONQUR FRAMEWORK

We now outline a specific instantiation of the CONQUR framework that can effectively navigate the large search space that arises in practical RL settings. We describe a heuristic, modified beam-search strategy with backtracking and priority scoring. Pseudocode is provided in Algorithm 2 (see Appendix B); here we simply outline some of the key refinements.

Our search process grows the tree in a breadth-first manner, and alternates between two phases. In an *expansion phase*, parent nodes are expanded, generating one or more child nodes with action assignments sampled from the Boltzmann distribution. For each child, we create target Q-labels, then optimize the child's regressor using consistency-penalized Bellman error (Eq. 2) as our loss. We thus forego strict policy consistency, and instead "encourage" consistency in regression. In a *dive phase*, each parent generates *one* child, whose action assignment is given by the usual max-actions selected by the parent node's regressor as in standard Q-learning. No additional diversity is considered in the dive phase, but consistency is promoted using consistency-penalized regression.

From the root, the search begins with an expansion phase to create $c$ children—$c$ is the *splitting factor*. Each child inherits its parent's consistency buffer from which we add the new action assignments that were used to generate that child's Q-labels. To limit the size of the tree, we only track a subset of the children, the *frontier* nodes, selected using one of several possible scoring functions. We select the top $\ell$-nodes for expansion, proceed to a dive phase and iterate.

It is possible to move beyond this "beam-like" approach and consider backtracking strategies that will return to unexpanded nodes at shallower depths of the tree. We consider this below as well.

### 3.6 RELATED WORK

Other work has considered multiple hypothesis tracking in RL. One particularly direct approach has been to use ensembling, where multiple Q-approximators are updated in parallel (Faußer & Schwenker, 2015; Osband et al., 2016; Anschel et al., 2017) then combined straightforwardly to reduce instability and variance. An alternative approach has been to consider population-based methods inspired by evolutionary search. For example, Conti et al. (2018) combine a novelty-search and quality diversity technique to improve hypothesis diversity and quality in RL. Khadka & Tumer (2018) consider augmenting an off-policy RL method with diversified population information from an evolutionary algorithm. Although these techniques do offer some benefit, they do not systematically target an identified weakness of Q-learning, such as delusion.

## 4 EMPIRICAL RESULTS

We experiment using the Atari test suite (Bellemare et al., 2013) to assess the performance of CONQUR. We first assess the impact of using the consistency penalty in isolation (without search) as a "regularizer" that promotes consistency with both DQN and DDQN. We then test the modified beam search described in Appendix B to assess the full power of CONQUR.

### 4.1 IMPACT OF CONSISTENCY PENALIZATION

We first study the effects of introducing the soft-policy consistency in isolation, augmenting both DQN and DDQN with the consistency penalty term. We train our models using an open-source implementation (Guadarrama et al., 2018) of both DQN and DDQN (with the same hyperparameters). We call these modified algorithms DQN($\lambda$) and DDQN($\lambda$), respectively, where $\lambda$ is the penalty coefficient defined in Eq. 2. Note that $\lambda = 0$ recovers the original methods. This is a lightweight

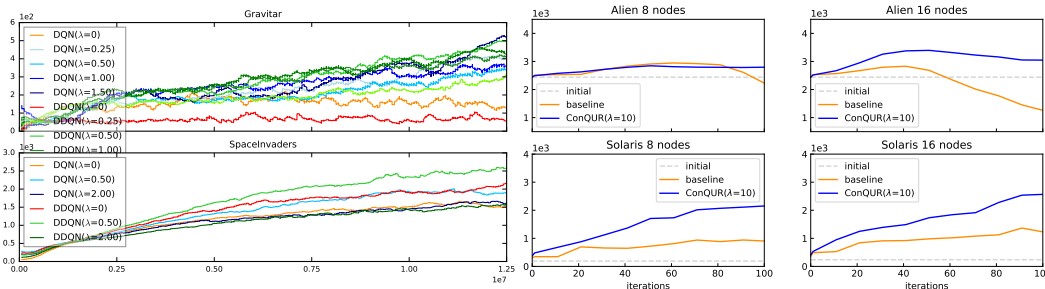

Fig. 2: Varying penalization $\lambda$ (no search procedure).

Fig. 3: Effects of increasing the number of nodes.

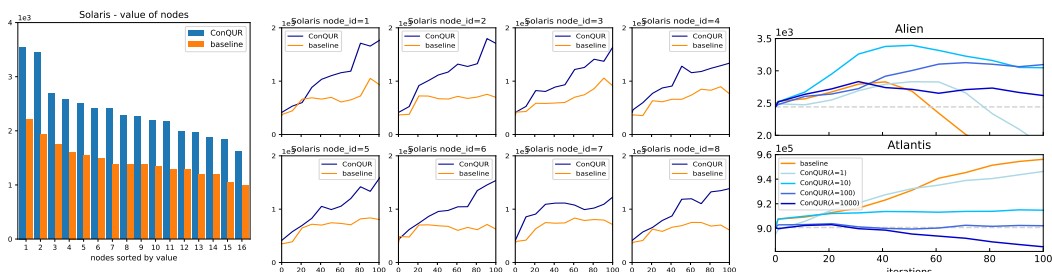

Fig. 4: Policy value of nodes, sorted.

Fig. 5: Training curves of 8 (of 16) sorted nodes.

Fig. 6: Effects of $\lambda$.

modification that can be applied readily to any regression-based Q-learning method, and serves to demonstrate the effectiveness of soft-policy consistency penalty. Since we don't consider search (i.e., don't track multiple hypotheses), we maintain a small consistency buffer using only the current data batch by sampling from the replay buffer—this prevents getting "trapped" by premature policy constraints. As the action assignment is the maximizing action of some network, $\sigma(s')$ can be computed easily for each batch. This results in a simple algorithmic extension that adds only an additional penalty term to the original TD-loss.

We train and evaluate DQN($\lambda$) and DDQN($\lambda$) for $\lambda = \{0.25, 0.5, 1, 1.5, 2\}$ on 19 Atari games. In training, $\lambda$ is initialized to 0 and slowly annealed to the desired value to avoid premature commitment to poor action assignments. Without annealing, the model tends fit to poorly informed action assignments during early phases of training, and thus fails to learn a good model.

The best $\lambda$ is generally different across games, depending on the nature of the game and the extent of delusional bias. Though a single $\lambda = 0.5$ works well across all games tested, Fig. 2 illustrates the effect of increasing $\lambda$ on two games. In Gravitar, increasing $\lambda$ generally results in better performance for both DQN and DDQN, whereas in SpaceInvaders, $\lambda = 0.5$ gives improvement over both baselines, but performance starts to degrade for $\lambda = 2$.

We compare the performance of the algorithms using each $\lambda$ value separately, as well as using the best $\lambda$ for each game. Under the best $\lambda$, DQN($\lambda$) and DDQN($\lambda$) outperform their "potentially delusional" counterparts on all except 3 and 2 games, respectively. In 9 of these games, each of DQN($\lambda$) and DDQN($\lambda$) beats *both* baselines. With a constant $\lambda = 0.5$, each algorithm still beats their respective baseline in 11 games. These results suggest that consistency penalization (independent of the general CONQUR model) can improve the performance of DQN and DDQN by addressing the delusional bias that is critical to learning a good policy. Moreover, we see that consistency penalization seems to have a different effect on learned Q-values than double Q-learning, which addresses maximization bias. Indeed, consistency penalization, when applied to DQN, can achieve gains that are greater than DDQN (in 15 games). Third, in 9 games DDQN($\lambda$) provides additional performance gains over DQN($\lambda$).

A detailed description of the experiments and further results can be found in Appendix C.

|  | **Rollouts** | **Bellman + Consistency Penalty** |
|---|---|---|
| BattleZone | 33796.30 | 32618.18 |
| BeamRider | 9914.00 | 10341.20 |
| Boxing | 83.34 | 83.03 |
| Breakout | 379.21 | 393.00 |
| MsPacman | 5947.78 | 5365.06 |
| Seaquest | 2848.04 | 3000.78 |
| SpaceInvader | 3442.31 | 3632.25 |
| StarGunner | 55800.00 | 56695.35 |
| Zaxxon | 11064.00 | 10473.08 |

Table 1: Results of CONQUR with 8 (split 2) nodes on 9 games using the proposed scoring function compared to evaluation using rollouts.

## 4.2 CONQUR RESULTS

We test the full CONQUR framework using the modified beam search discussed above. Rather than training a full Q-network, for effective testing of its core principles, we leverage pre-trained networks from the Dopamine package Castro et al. (2018).[4]. These networks have the same architecture as in Mnih et al. (2015) and are trained on 200M frames with sticky actions using DQN. We use CONQUR to retrain only the last (fully connected) layer (implicitly freezing the other layers), which can be viewed as a linear Q-approximator over the features learned by the CNN. We run CONQUR using only 4M addtional frames to train our Q-regressors.[5]

We consider splitting factors $c$ of $2$ and $4$; impose a limit on the frontier size of 8 or 16; and an expansion factor of 2 or 4. The dive phase is always of length 9 (i.e., 9 batches of data), giving an expansion phase every 10 iterations. Regressors are trained using the loss in Eq. 2 and the consistency buffer comprises *all* prior action assignments. (See Appendix D for details, hyperparameter choices and more results.)

We run CONQUR with $\lambda = \{1, 10, 100, 1000\}$ and select the best performing policy. We initially test two scoring approaches, policy evaluation using rollouts and scoring using the loss function (Bellman error with consistency penalty). Results comparing the two on a small selection of games are shown in Table 1. While rollouts, not surprisingly, tend to give rise to better-performing policies, consistent-Bellman scoring is competitive. Since the latter much less computationally intense, and does not require sampling the environment, we use it throughout our remaining experiments.

We compare CONQUR with the value of the pre-trained DQN. We also evaluate a "multi-DQN" baseline that applies multiple versions of DQN independently, warm-starting from the same pre-trained DQN. It uses the same number of frontier nodes as CONQUR, and is otherwise trained identically as CONQUR but with direct Bellman error (no consistency penalty). This gives DQN the same advantage of multiple-hypothesis tracking as CONQUR but without policy consistency.

We test on 59 games, comparing CONQUR with frontier size 16 and expansion factor 4 and splitting factor 4 (16-4-4) with backtracking (as described in the Appendix D) resulted in significant improvements to the pre-trained DQN, with an average score improvement of 125% (excluding games with non-positive pre-trained score). The only games without improvement are Montezuma's Revenge, Tennis, PrivateEye and BankHeist. This demonstrates that, even when simply retraining the last layer of a highly tuned DQN network, removing delusional bias has the potential to offer strong improvements in policy performance. It is able exploit the reduced parameterization to obtain these gains with only 4M frames of training data. Roughly, a half-dozen games have outsized score improvements, including Solaris (11 times greater value), Tutankham (6.5 times) and WizardOfWor (5 times).[6]

Compared to the stronger multi-DQN baseline (with 16 nodes), CONQUR wins by at least a 10% margin in 20 games, while 22 games see improvements of 1–10% and 8 games show little effect

---

[4]See `https://github.com/google/dopamine`

[5]This approach is simply to reduce the computational and memory footprint of our experiments. The framework is not limited to this approach.

[6]This may be in part, but not fully, due to the sticky-action training of the pre-trained model.

(plus/minus 1%) and 7 games show a decline of greater than 1% (most are 1–6% with the exception of Centipede at -12% and IceHockey at -86%). Results are similar when comparing CONQUR and multi-DQN each with 8 nodes (8-2-2): 9 games exhibit 10%+ improvement, 21 games show 1–8% improvement, 12 games perform comparably and 7 games do worse under CONQUR. A table of complete results appears in Appendix D.3, Table 4, and training curves (all games, all $\lambda$) in Fig. 11.

Increasing the number of nodes from 8 to 16 generally leads to better performance for CONQUR, with 38 games achieving strictly higher scores with 16 nodes (16-4-4): 16 games with 10%+ improvement, 5 games tied and the remaining 16 games performing worse (only a few with a 5%+ decline). Fig. 3 shows the (smoothed) effect of increasing the number of nodes for a fixed $\lambda = 10$. The $y$-axis represents the rollout value of the best frontier node (i.e., the greedy policy of its Q-regressor) as a function of the training iteration. For both Alien and Solaris, the multi-DQN (baseline) training curve is similar with both 8 and 16 nodes, but CONQUR improves Alien from 3k to 4.3k while Solaris improves from 2.2k to 3.5k.

Fig. 4 and Fig. 5 (smoothed, best frontier node) shows node policy values and training curves, respectively, for Solaris. When considering nodes ranked by their policy value, comparing nodes of equal rank generated by CONQUR and by multi-DQN (baseline), we see that CONQUR nodes dominate their multi-DQN counterparts: the three highest-ranked nodes achieve a score improvement of 18%, 13% and 15%, respectively, while the remaining nodes achieve improvements of roughly 11–12%. Fig. 6 (smoothed, best frontier node) shows the effects of varying $\lambda$. In Alien, increasing $\lambda$ from 1 to 10 improves performance, but it starts to decline for higher values of 100 and 1000. This is similar to patterns observed in 4.1 and represents a trade-off between emphasizing consistency and not over-committing to action assignments. In Atlantis, stronger penalization tends to degrade performance. In fact, the stronger the penalization, the worse the performance.

## 5 CONCLUDING REMARKS

We have introduced CONQUR, a framework for mitigating delusional bias in value-based RL that relaxes some of the strict assumptions of exact delusion-free algorithms to ensure scalability. Its two main components are (a) a tree-search procedure used to create and maintain diverse, promising Q-regressors (and corresponding information sets); and (b) a consistency penalty that encourages "maximizing" actions to be consistent with the FA class. CONQUR embodies elements of both value-based and policy-based RL: it can be viewed as using partial policy constraints to bias the value estimator or as a means of using candidate value functions to bias the search through policy space. Empirically, we find that CONQUR can improve the quality of existing approximators by removing delusional bias. Moreover, the consistency penalty applied on its own, directly in DQN or DDQN, itself can improve the quality of the induced policies.

Given the generality of the CONQUR framework, there remain numerous interesting directions for future research. Other methods for nudging regressors to be policy-consistent include exact consistency (constrained regression), other regularization schemes that bias the regressor to fall within the information set, etc. Given its flexibility, more extensive exploration of search strategies (e.g., best first), child-generation strategies, and node scoring schemes should be examined within CONQUR. Our (full) experiments should also be extended beyond those that warm-start from a DQN model, as should testing CONQUR in other domains.

Other connections and generalizations are of interest as well. We believe our methods can be extended to both continuous actions and soft max-action policies. We suspect that there is a connection between maintaining multiple "hypotheses" (i.e., Q-regressors) and notions in distributional RL, which maintains distributions over action values Bellemare et al. (2017).

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

## A    AN EXAMPLE OF DELUSIONAL BIAS

We describe an example, taken directly from (Lu et al., 2018), to show concretely how delusional bias causes problems for Q-learning with function approximation. The MDP in Fig. 7 illustrates the phenomenon: Lu et al. (2018) use a linear approximator over a specific set of features in this MDP to show that:

(a) No $\pi \in G(\Theta)$ can express the optimal (unconstrained) policy (which requires taking $a_2$ at each state);

(b) The optimal *feasible* policy in $G(\Theta)$ takes $a_1$ at $s_1$ and $a_2$ at $s_4$ (achieving a value of $0.5$).

(c) Online Q-learning (Eq. 1) with data generated using an $\varepsilon$-greedy behavior policy must converge to a fixed point (under a range of rewards and discounts) corresponding to a "compromise" admissible policy which takes $a_1$ at both $s_1$ and $s_4$ (value of $0.3$).

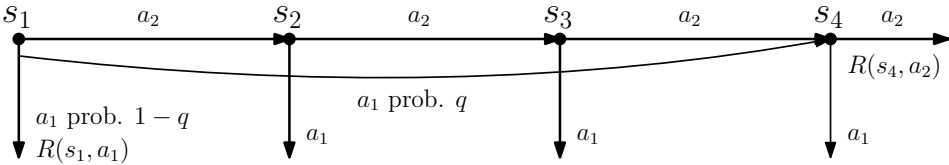

Fig. 7: A simple MDP (Lu et al., 2018).

---

**Algorithm 1** CONQUR SEARCH (Generic, depth-first)

---

**Input:** Data sets $D_k, D_{k+1}, \ldots D_T$; regressor $\hat{Q}_{k-1}$; and assignment $\sigma$ over $D_{\leq k-1} = \cup_{1 \leq j \leq k-1} D_j$ reflecting prior data; policy class $\Theta$.
1: Let $\Sigma_{\Theta,\sigma} = \{\sigma_k \in \Sigma_\Theta(D_j) : \sigma_k \cup \sigma \text{ is consistent}\}$
2: **for all** $\sigma_k^j \in \Sigma_{\Theta,\sigma}$ **do**
3:     Training set $S \leftarrow \{\}$
4:     **for all** $(s, a, r, s') \in D_k$ **do**
5:         $q \leftarrow r + \gamma \hat{Q}_{k-1}(s', \sigma_k^j(s'))$
6:         $S \leftarrow S \cup \{((s, a), q)\}$
7:     **end for**
8:     Train $\hat{Q}_k^j$ using training set $S$
9:     **if** $k = T$ **then**
10:        Return $\hat{Q}_k^j$ // terminate
11:     **else**
12:        Return SEARCH($D_{k+1}, \ldots D_T$; $\hat{Q}_k^j$; $\sigma_k^j \cup \sigma$; $\Theta$) // recurse
13:     **end if**
14: **end for**

---

Q-learning fails to find a reasonable fixed-point because of delusion. Consider the backups at $(s_2, a_2)$ and $(s_3, a_2)$. Suppose $\hat{\theta}$ assigns a "high" value to $(s_3, a_2)$, so that $Q_{\hat{\theta}}(s_3, a_2) > Q_{\hat{\theta}}(s_3, a_1)$ as required by $\pi_{\theta^*}$. They show that any such $\hat{\theta}$ also accords a "high" value to $(s_2, a_2)$. But $Q_{\hat{\theta}}(s_2, a_2) > Q_{\hat{\theta}}(s_2, a_1)$ is inconsistent the first requirement. As such, any update that makes the Q-value of $(s_2, a_2)$ higher *undercuts the justification* for it to be higher (i.e., makes the "max" value of its successor state $(s_3, a_2)$ lower). This occurs not due to approximation error, but the inability of Q-learning to find the value of the optimal *representable* policy.

# B  ALGORITHMS

The pseudocode of (depth-first) version of the CONQUR search framework is listed in Algorithm 1. As discussed in Sec. 3.5, a more specific instantiation of the CONQUR algorithm is listed in Algorithm. 2.

# C  ADDITIONAL DETAIL: EFFECTS OF CONSISTENCY PENALIZATION

## C.1  DELUSIONAL BIAS IN DQN AND DDQN

Both DQN and DDQN uses a delayed version of the $Q$-network $Q_{\theta^-}(s', a')$ for label generation, but in a different way. In DQN, $Q_{\theta^-}(s', a')$ is used for both value estimate and action assignment $\sigma_{\text{DQN}}(s') = \text{argmax}_{a'} Q_{\theta_k}(s', a')$, whereas in DDQN, $Q_{\theta^-}(s', a')$ is used only for value estimate and the action assignment is computed from the current network $\sigma_{\text{DDQN}}(s') = \text{argmax}_{a'} Q_{\theta_k}(s', a')$.

With respect to delusional bias, action assignment of DQN is consistent for all batches after the latest network weight transfer, as $\sigma_{\text{DQN}}(s')$ is computed from the same $Q_{\theta^-}(s', a')$ network. DDQN, on the other hand, could have very inconsistent assignments, since the action is computed from the current network that is being updated at every step.

---

**Algorithm 2** Modified Beam Search Instantiation of CONQUR Algorithm

---

**Input:** Search control parameters: $m, \ell, c, d, T$
1: Maintain list of data batches $D_1, ..., D_k$, initialized empty
2: Maintain candidate pool $P$ of at most $m$ nodes, initialized $P = \{n_0\}$
3: Maintain frontier list $F$ of $\ell^c$ nodes
4: Maintain for each node $n_k^i$ a regressor $\theta_k^i$ and an ancestor assignment $\sigma_k^i$

5: **for** each search level $k \leq T$ **do**
6:     Find top scoring node $n^1 \in P$
7:     Use $\varepsilon$-greedy policy extracted from $Q_{\theta^1}$ to collect next data batch $D_k$

8:     **if** $k$ is an expansion level **then**
9:         Select top $\ell$ scoring nodes $n^1, ..., n^\ell \in P$
10:         **for** each selected node $n^i$ **do**
11:             Generate $c$ children $n^{i,1}, ..., n^{i,c}$ using Boltzmann sampling on $D_k$ with $Q_{\theta^i}$
12:             **for** each child $n^{i,j}$ **do**
13:                 Let assignment history $\sigma^{i,j}$ be $\sigma^i \cup \{new\ assignment\}$
14:                 Determine regressor $\theta^{i,j}$ by applying update (3) from $\theta^i$
15:             **end for**
16:             Score and add child nodes to the candidate pool $P$
17:             Assign frontier nodes to set of child nodes, $F = \{n^{i,j}\}$
18:             **if** $|P| > m$ **then**
19:                 evict bottom scoring nodes, keeping top $m$ in $P$
20:             **end if**
21:         **end for**
22:     **end if**

23:     **if** $k$ is a refinement ("dive") level **then**
24:         **for** each frontier node $n^{i,j} \in F$ **do**
25:             Update regressor $\theta^{i,j}$ by applying update (3) to $\theta^{i,j}$
26:         **end for**
27:     **end if**

28:     Run $d$ "dive" levels after each expansion level

29: **end for**

---

## C.2 TRAINING METHODOLOGY AND HYPERPARAMETERS

We implement consistency penalty on top of the DQN and DDQN algorithm by modifying the open-source TF-Agents library (Guadarrama et al., 2018). In particular, we modify existing `DqnAgent` and `DdqnAgent` by adding a consistency penalty term to the original TD loss.

We use TF-Agents implementation of DQN training on Atari with the default hyperparameters, which are mostly the same as that used in the original DQN paper (Mnih et al., 2015). For conveniece to the reader, some important hyperparameters are listed in Table 2. The reward is clipped between $[-1, 1]$ following the original DQN.

## C.3 EVALUATION METHODOLOGY

We empirically evaluate our modified DQN and DDQN agents trained with consistency penalty on 15 Atari games. Evaluation is run using the training and evaluation framework for Atari provided in TF-Agents without any modifications.

## C.4 DETAILED RESULTS

Fig. 8 shows the effects of varying $\lambda$ on both DQN and DDQN. Table 3 summarizes the best penalties for each game and their corresponding scores. Fig. 9 shows the training curves of the best penalization constants. Finally, Fig. 10 shows the training curves for a fixed penalization of $\lambda = 1/2$. The datapoints in each plot of the aforementioned figures are obtained by taking windows of size 30

| Hyper-parameter | Value |
|---|---|
| Mini-batch size | 32 |
| Replay buffer capacity | 1 million transitions |
| Discount factor $\gamma$ | 0.99 |
| Optimizer | RMSProp |
| Learning rate | 0.00025 |
| Convolution channel | $32, 64, 64$ |
| Convolution filter size | $(8 \times 8), (4 \times 4), (3 \times 3)$ |
| Convolution stride | 4, 2, 1 |
| Fully-connected hidden units | 512 |
| Train exploration $\varepsilon_{\text{train}}$ | 0.01 |
| Eval exploration $\varepsilon_{\text{eval}}$ | 0.001 |

Table 2: Hyperparameters for training DQN and DDQN with consistency penalty on Atari.

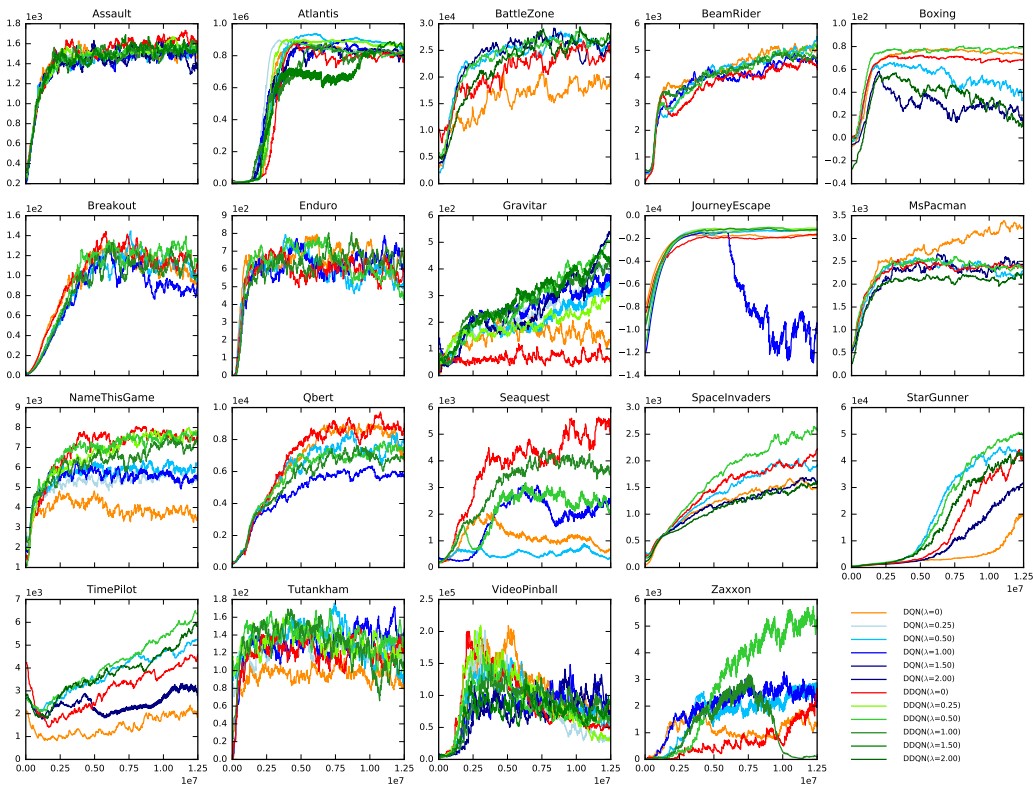

Fig. 8: DQN and DDQN training curves for different penalty constant $\lambda$.

steps, and within each window, we take the largest policy value (and over $\approx$2–5 multiple runs). This is done to reduce visual clutter.

## D  ADDITIONAL DETAIL: CONQUR RESULTS

Our results use a frontier queue of size ($F$) 8 or 16 (these are the top scoring leaf nodes which receive gradient updates and rollout evaluations during training). To generate training batches, we select the best node's regressor according to our scoring function, from which we generate training samples (transitions) using $\varepsilon$-greedy. Results are reported in Table 4 and 5, and related figures where max number of nodes are 8 or 16. We used Bellman error plus consistency penalty as our scoring function. During the training process, we also calibrated the scoring to account for the depth difference between

| | DQN | $\lambda_{best}$ | DQN($\lambda_{best}$) | DDQN | $\lambda'_{best}$ | DDQN($\lambda'_{best}$) |
|---|---|---|---|---|---|---|
| Assault | 2546.56 | 1.5 | **3451.07** | 2770.26 | 1 | 2985.74 |
| Atlantis | 995460.00 | 0.5 | **1003600.00** | 940080.00 | 1.5 | 999680.00 |
| BattleZone | **67500.00** | 2 | 55257.14 | 47025.00 | 2 | 48947.37 |
| BeamRider | 7124.90 | 0.5 | **7216.14** | 5926.59 | 0.5 | 6784.97 |
| Boxing | 86.76 | 0.5 | 90.01 | 82.80 | 0.5 | **91.29** |
| Breakout | 220.00 | 0.5 | 219.15 | 214.25 | 0.5 | **242.73** |
| Enduro | 1206.22 | 0.5 | **1430.38** | 1160.44 | 1 | 1287.50 |
| Gravitar | 475.00 | 1.5 | **685.76** | 462.94 | 1.5 | 679.33 |
| JourneyEscape | -1020.59 | 0.25 | -696.47 | -794.71 | 1 | **-692.35** |
| MsPacman | **4104.59** | 2 | 4072.12 | 3859.64 | 0.5 | 4008.91 |
| NameThisGame | 7230.71 | 1 | 9013.48 | 9618.18 | 0.5 | **10210.00** |
| Qbert | 13270.64 | 0.5 | **14111.11** | 13388.92 | 1 | 12884.74 |
| Seaquest | 5849.80 | 1 | 6123.72 | **12062.50** | 1 | 7969.77 |
| SpaceInvaders | 2389.22 | 0.5 | 2707.83 | 3007.72 | 0.5 | **4080.57** |
| StarGunner | 40393.75 | 0.5 | 55931.71 | 55957.89 | 0.5 | **60035.90** |
| TimePilot | 4205.83 | 2 | 7612.50 | 6654.44 | 2 | **7964.10** |
| Tutankham | 222.76 | 1 | **265.86** | 243.20 | 0.25 | 247.17 |
| VideoPinball | **569502.19** | 0.25 | 552456.00 | 509373.50 | 0.25 | 562961.50 |
| Zaxxon | 5533.33 | 1 | **10520.00** | 7786.00 | 0.5 | 10333.33 |

Table 3: Consistency penalty ablation results on best penalty constants for DQN and DDQN.

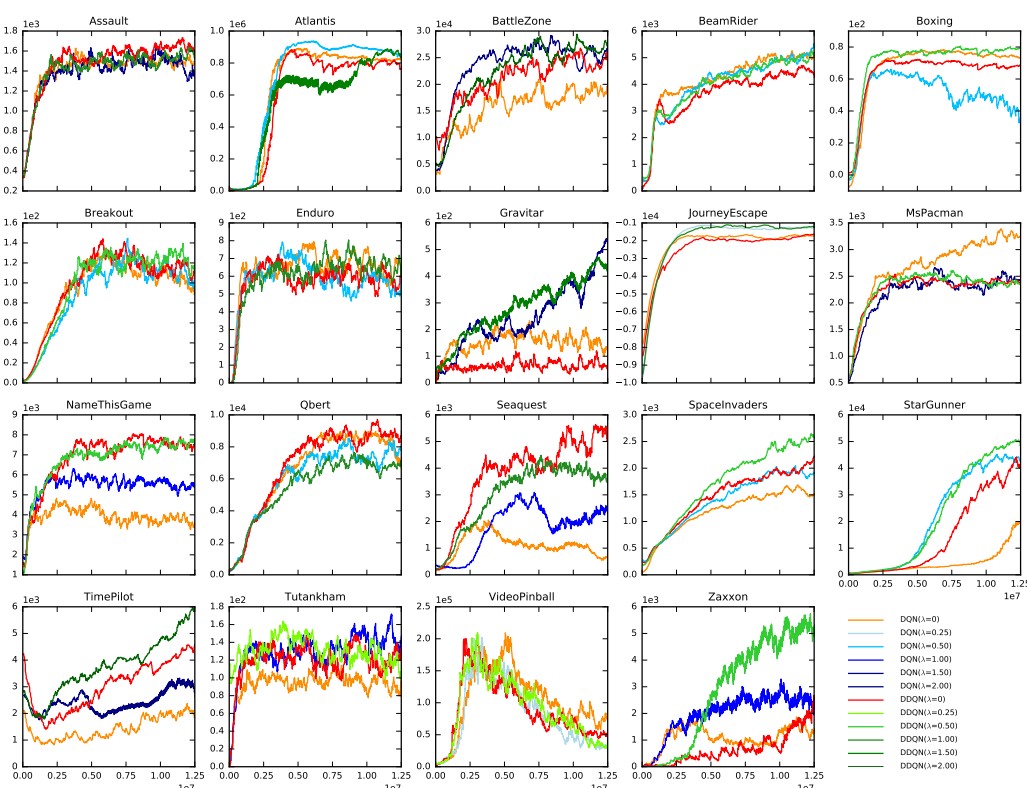

Fig. 9: DQN and DDQN training curves for the respective best $\lambda$ and baseline.

the leaf nodes at the frontier versus the leaf nodes in the candidate pool. We calibrated by taking the mean of the difference between scores of the current nodes in the frontier with their parents. We scaled this difference by multiplying with a constant of 2.5.

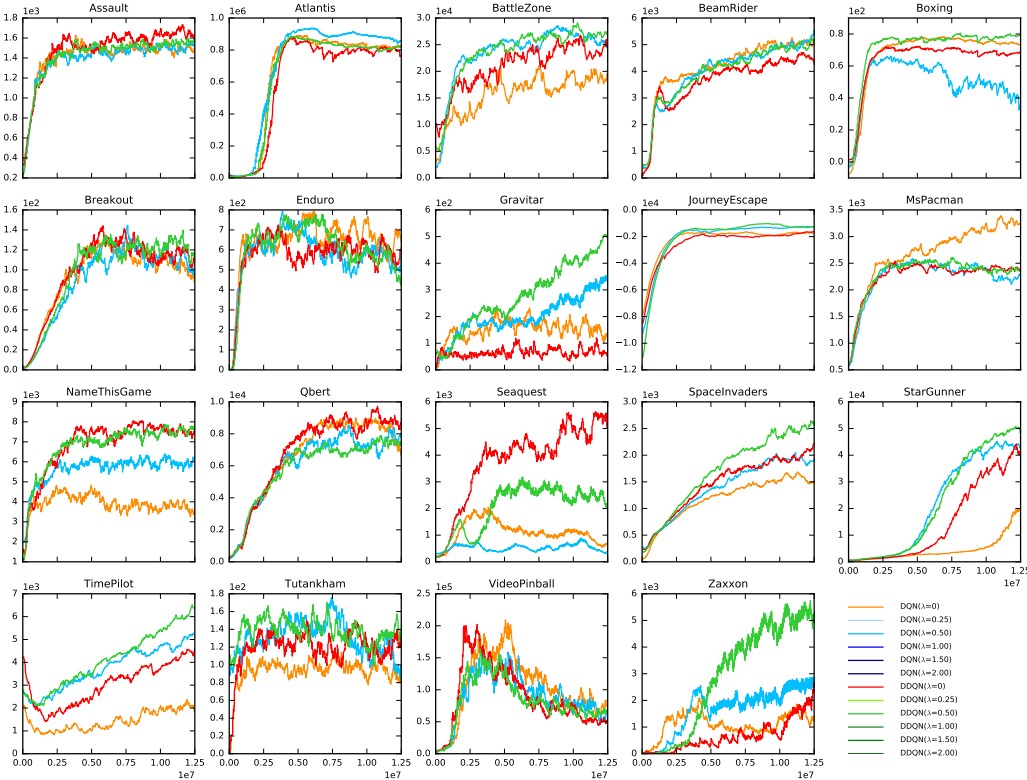

Fig. 10: DQN and DDQN training curves for $\lambda = 0.5$ and the baseline.

In our implementation, we initialized our Q-network with a pre-trained DQN. We start with the expansion phase. During this phase, each parent node splits into $l$ children nodes and the Q-labels are generated using action assignments from the Boltzmann sampling procedure, in order to create high quality and diversified children. We start the dive phase until the number of children generated is at least $F$. In particular, with $F = 16$ configuration, we performed the expansion phase at the zero-th and first iterations, and then at every tenth iteration starting at iteration 10, then at 20, and so on until ending at iteration 90. In the $F = 8$ configuration, the expansion phase occurred at the zero-th and first iterations, then at every tenth iterations starting at iterations 10 and 11, then at iterations 20 and 21, and so on until ending at iterations 90 and 91. All other iterations execute the "dive" phase. For every fifth iteration, Q-labels are generated from action assignments sampled according to the Boltzmann distribution. For all other iterations, Q-labels are generated in the same fashion as the standard Q-learning (taking the max Q-value). The generated Q-labels along with the consistency penalty are then converted into gradient updates that applies to one or more generated children nodes.

### D.1 TRAINING METHODOLOGY AND HYPERPARAMETERS

Each iteration consists of 10k transitions sampled from the environment. Our entire training process has 100 iterations which consumes 1M transitions or 4M frames. We used RMSProp as the optimizer with a learning rate of $2.5 \times 10^{-6}$. One training iteration has 2.5k gradient updates and we used a batch size of 32. We replace the target network with the online network every fifth iteration and reward is clipped between $[-1, 1]$. We use a discount value of $\gamma = 0.99$ and $\varepsilon$-greedy with $\varepsilon = 0.01$ for exploration. Details of hyper-parameter settings can be found in Table 6 (for 16 nodes) and Table 7 (for 8 nodes).

## D.2 EVALUATION METHODOLOGY

We empirically evaluate our algorithms on 59 Atari games (Bellemare et al., 2013), and followed the evaluation procedure as in Hasselt et al. (2016). We evaluate our agents every 10th iterations (and also the initial and first iteration) by suspending our training process. We evaluate on 500k frames, and we cap the length of the episodes for 108k frames. We used $\varepsilon$-greedy as the evaluation policy with $\varepsilon = 0.001$.

## D.3 DETAILED RESULTS

Fig. 11 shows training curves of CONQUR with 16 nodes under different penalization strengths $\lambda$. Each plotted step of each training curve (including the baseline) shows the best performing node's policy value as evaluated with full rollouts. Table 4 shows the summary of the highest policy values achieved for all 59 games for CONQUR and the baseline under 8 and 16 nodes. Table 5 shows a similar summary, but without no-op starts (i.e. policy actions are applied immediately). Both the baseline and CONQUR improve overall, but CONQUR's advantage over the baseline is amplified. This may suggest that for more deterministic MDP environments, CONQUR may have even better improvements. The results on 16 and 8 nodes use a splitting factor of 4 and 2, respectively.

| | CONQUR (8 nodes) | CONQUR (16 nodes) | Baseline (8 nodes) | Baseline (16 nodes) | Checkpoint |
|---|---|---|---|---|---|
| AirRaid | 11365.00 | **11627.68** | 9578.13 | 9565.15 | 6962.21 |
| Alien | 3585.00 | **4340.42** | 3327.33 | 3213.07 | 2440.52 |
| Amidar | 552.90 | 632.63 | 655.54 | **668.47** | 207.08 |
| Assault | 2005.68 | 1971.44 | 2002.59 | **2050.48** | 1862.83 |
| Asterix | 6094.21 | **6395.56** | 5589.39 | 4984.82 | 2476.85 |
| Asteroids | 1313.21 | **1395.48** | 1224.42 | 1366.25 | 686.19 |
| Atlantis | 952920.00 | 957120.00 | 966080 | **974880.00** | 900740.00 |
| BankHeist | 873.18 | 873.18 | 894.25 | **897.09** | 873.18 |
| BattleZone | 32618.18 | **35178.57** | 30966.1 | 34796.30 | 26000.00 |
| BeamRider | **10341.20** | 9426.00 | 8643.85 | 10021.75 | 6131.14 |
| Berzerk | 716.00 | **797.60** | 630.8 | 637.37 | 531.26 |
| Bowling | 33.26 | **47.86** | 25.41 | 43.64 | 25.03 |
| Boxing | 83.03 | **84.21** | 82.39 | 82.57 | 80.12 |
| Breakout | 393.00 | **394.31** | 354.31 | 367.17 | 326.83 |
| Carnival | 5080.02 | **5121.81** | 4921.79 | 4907.37 | 4725.68 |
| Centipede | 2773.36 | 3914.76 | 4305.33 | **4383.07** | 745.20 |
| ChopperCommand | 7683.89 | **13812.50** | 6971.43 | 8866.67 | 2500.00 |
| CrazyClimber | **141784.00** | 139857.14 | 138336.84 | 125383.34 | 68222.22 |
| DemonAttack | 13270.77 | **14469.40** | 11544.17 | 11496.00 | 6436.40 |
| DoubleDunk | -10.49 | **-9.58** | -14.78 | -14.89 | -17.14 |
| ElevatorAction | 120.69 | 100.00 | **180** | 83.33 | 0.00 |
| Enduro | 938.64 | **1056.00** | 1006.18 | 1002.70 | 566.56 |
| FishingDerby | **21.85** | 14.59 | 14.45 | 14.32 | 11.88 |
| Freeway | 32.60 | 32.63 | **32.71** | 32.65 | 32.47 |
| Frostbite | 317.78 | **333.25** | 220.8 | 224.29 | 167.21 |
| Gopher | **12782.42** | 12780.00 | 7948.5 | 8436.00 | 5066.69 |
| Gravitar | 467.65 | 448.04 | **483.71** | 460.06 | 399.72 |
| Hero | 20797.14 | **20816.77** | 20803.64 | 20799.20 | 20612.46 |
| IceHockey | -2.98 | -3.40 | -3.07 | **-1.83** | -8.49 |
| Jamesbond | **876.12** | 725.76 | 727.54 | 704.48 | 626.12 |
| JourneyEscape | 1622.41 | **3094.22** | 1523.12 | 2798.84 | -1099.43 |
| Kangaroo | 12564.87 | **13717.65** | 10979.07 | 10972.09 | 10629.55 |
| Krull | 9496.24 | **9803.63** | 9239.06 | 9443.49 | 4002.90 |
| MontezumaRevenge | 0.00 | 0.00 | 0.00 | 0.00 | 0.00 |
| MsPacman | 5365.06 | **5697.31** | 5108.51 | 5658.89 | 4185.28 |
| NameThisGame | 9220.43 | **9462.17** | 9024.78 | 8944.78 | 5497.81 |
| Phoenix | 5581.67 | 5486.00 | 5472.22 | **5624.29** | 4645.71 |
| Pitfall | 0.00 | 0.00 | 0.00 | 0.00 | -20.00 |
| Pong | 21.00 | 21.00 | 21.00 | 21.00 | 20.92 |
| Pooyan | 6636.03 | **6688.46** | 5358.33 | 5444.67 | 4551.30 |
| PrivateEye | 100.00 | 100.00 | 100.00 | 100 | 100.00 |
| Qbert | **17637.16** | 15751.71 | 15239.14 | 15319.86 | 8626.09 |
| Riverraid | 15434.68 | **16388.39** | 14743.12 | 15570.32 | 10808.57 |
| RoadRunner | **51097.73** | 50872.34 | 48465.45 | 47603.45 | 47603.45 |
| Robotank | **64.59** | 64.56 | 60.95 | 62.94 | 50.19 |
| Seaquest | 3000.78 | **3399.02** | 2416.07 | 3110.88 | 1050.27 |
| Skiing | -9409.16 | **-9282.93** | -9315.58 | -9456.21 | -29911.07 |
| Solaris | 3013.33 | **3548.89** | 1917.14 | 2208.89 | 293.96 |
| SpaceInvaders | 3632.25 | **3873.55** | 3658.81 | 3520.00 | 2814.57 |
| StarGunner | 56695.35 | **58729.27** | 53477.78 | 55084.09 | 53477.78 |
| Tennis | **0.20** | 0.00 | 0.00 | 0.00 | 0.00 |
| TimePilot | 7755.89 | **10743.18** | 7118.33 | 5043.21 | 3833.54 |
| Tutankham | 257.00 | **292.00** | 231.63 | 233.83 | 39.07 |
| UpNDown | 29612.31 | **32601.54** | 28862.76 | 31623.93 | 5789.64 |
| Venture | 202.04 | **254.55** | 35.29 | 22.86 | 0.00 |
| VideoPinball | **501510.50** | 480730.59 | 325911.72 | 434650.41 | 283719.04 |
| WizardOfWor | 9932.61 | **12197.73** | 5747.37 | 8771.11 | 2013.24 |
| YarsRevenge | 26152.97 | 27801.24 | **28064.77** | 27007.31 | 25297.36 |
| Zaxxon | 10473.08 | 11154.17 | 10316.36 | **12422.00** | 5256.94 |

Table 4: Summary of scores with $\varepsilon$-greedy ($\varepsilon = 0.001$) evaluation with up to 30 no-op starts.

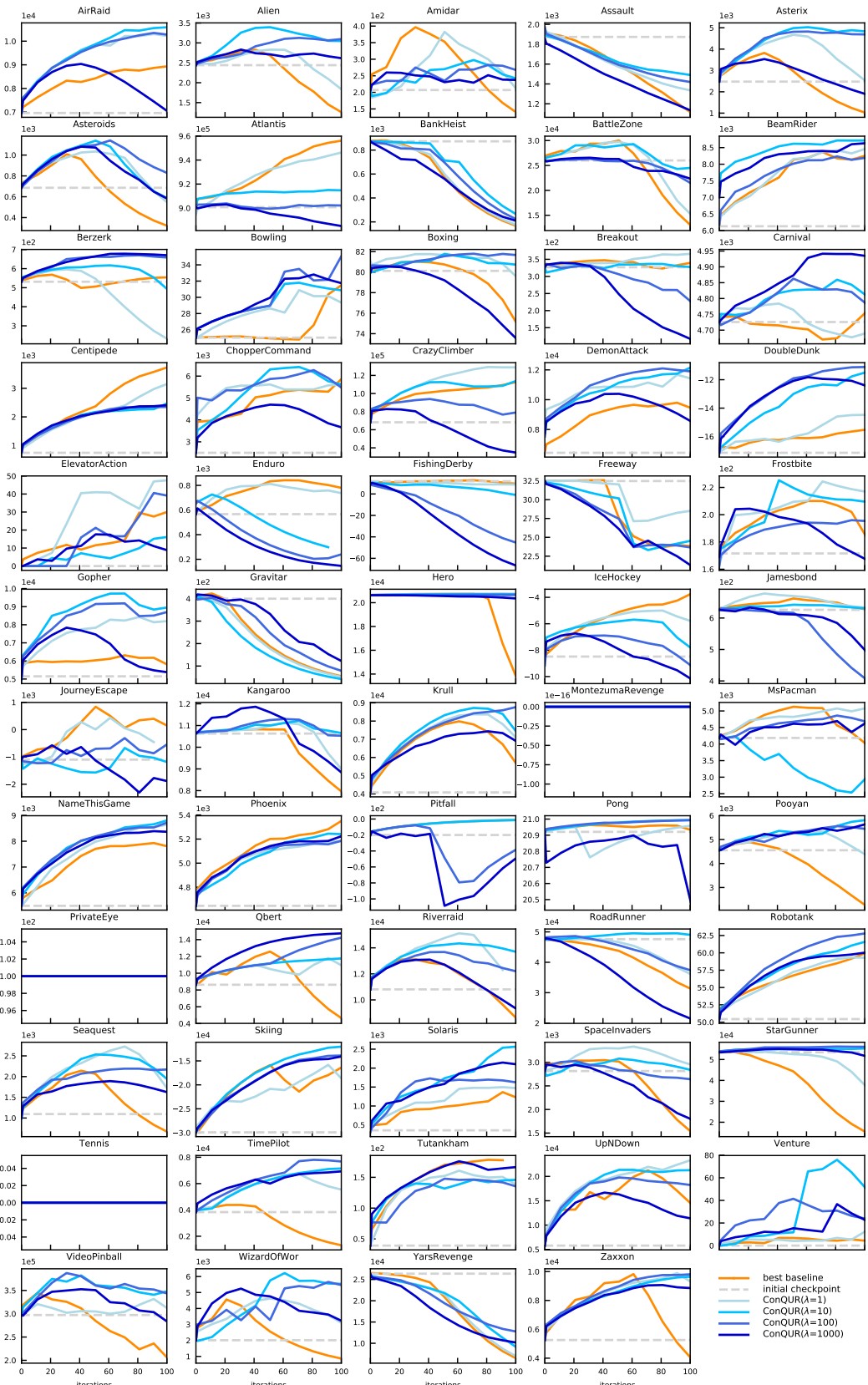

Fig. 11: Training curves on 16 nodes with up to 30 no-op starts.

|  | CONQUR(8 nodes) | Baseline (8 nodes) | Checkpoint |
|---|---|---|---|
| AirRaid | **21100.00** | 17200.00 | 4825.00 |
| Alien | **6480.00** | 3970.42 | 2990.00 |
| Amidar | 742.00 | **762.00** | 188.00 |
| Assault | 1872.77 | **1971.55** | 1678.14 |
| Asterix | 68300.00 | **7700.00** | 2605.97 |
| Asteroids | **2350.00** | 1180.00 | 430.22 |
| Atlantis | **969700.00** | 940300.00 | 922700.00 |
| BankHeist | 940.00 | **980.00** | 860.00 |
| BattleZone | **65000.00** | 48000.00 | 30000.00 |
| BeamRider | **16694.00** | 13012.11 | 9026.86 |
| Berzerk | **1223.33** | 850.48 | 450.00 |
| Bowling | **33.00** | 25.03 | 25.03 |
| Boxing | **98.39** | 90.38 | 87.25 |
| Breakout | **415.00** | 3.00 | 2.40 |
| Carnival | **5776.36** | 5331.79 | 4310.00 |
| Centipede | **6192.00** | 4486.31 | 1952.50 |
| ChopperCommand | **16544.19** | 6900.00 | 1197.14 |
| CrazyClimber | **156500.00** | 144500.00 | 22100.00 |
| DemonAttack | **35611.54** | 31101.00 | 7877.30 |
| DoubleDunk | **0.00** | -2.00 | -6.00 |
| ElevatorAction | 0.00 | 0.00 | 0.00 |
| Enduro | **1399.00** | 1065.00 | 197.95 |
| FishingDerby | **43.67** | 25.82 | 16.58 |
| Freeway | 33.00 | 33.00 | 33.00 |
| Frostbite | 220.00 | **449.88** | 112.47 |
| Gopher | **18040.00** | 11320.00 | 3720.00 |
| Gravitar | **850.00** | 700.00 | 598.84 |
| Hero | **20845.00** | 20780.44 | 20610.04 |
| IceHockey | **-0.77** | -1.97 | -9.10 |
| Jamesbond | 750.00 | 750.00 | 600.00 |
| JourneyEscape | **9300.00** | 1007.06 | -2687.65 |
| Kangaroo | **14700.00** | 10700.00 | 10700.00 |
| Krull | **9753.00** | 9459.00 | 687.92 |
| MsPacman | 5660.00 | **6310.00** | 4180.00 |
| NameThisGame | **12470.00** | 11670.00 | 4,750.00 |
| Phoenix | **5990.00** | 5770.00 | 4815.00 |
| Pitfall | 0.00 | 0.00 | 0.00 |
| Pong | 21.00 | 21.00 | 21.00 |
| Pooyan | **8910.00** | 7945.00 | 5320.00 |
| Qbert | **15650.00** | 15375.00 | 7221.43 |
| Riverraid | **20280.00** | 17000.00 | 7840.00 |
| RoadRunner | 63500.00 | 63500.00 | 42924.14 |
| Robotank | **75.00** | 72.00 | 52.65 |
| Skiing | **-9,044.00** | -30,000.00 | -30,000.00 |
| Solaris | **912.00** | 582.86 | 0.00 |
| StarGunner | **64900.00** | 61000.00 | 53800.00 |
| Tennis | 0.00 | 0.00 | -0.20 |
| TimePilot | **7937.10** | 4692.75 | 3801.84 |
| Tutankham | **252.00** | 203.00 | 30.00 |
| UpNDown | **71220.00** | 25470.00 | 4260.00 |
| Venture | **200.00** | 0.00 | 0.00 |
| VideoPinball | **778319.00** | 502803.00 | 23520.80 |
| WizardOfWor | **11700.00** | 4900.00 | 2000.00 |
| YarsRevenge | **34199.42** | 28605.70 | 13098.00 |
| Zaxxon | **17800.00** | 14300.00 | 5797.18 |

Table 5: Summary of scores with $\varepsilon$-greedy ($\varepsilon = 0.001$) evaluation, without no-op starts.

| Hyperparameters | Description | Value |
|---|---|---|
| Splitting factor $c$ | Controls the number of children created from a parent node | 4 |
| Candidate pool size $m$ | Pool of candidate leaf nodes for selection into the dive or expansion phase | 46 |
| Maximum frontier nodes $F$ | Maximum number of child leaf nodes for the dive phase | 16 |
| Top nodes to expand $l$ | Select the top $l$ nodes from the candidate pool for the expansion phase. | 4 |
| Dive levels $d$ to run | We run $d$ levels of diving phase after each expansion phase | 9 |
| Boltzmann iteration | Every module this number of iteration/level, Q-labels are generated from Boltzmann distribution in order to create diversified node. | 5 |
| Online network target network swap frequency | Iteration (Frequency) at which the online network parameters swap with the target network | 5 |
| Evaluation frequency | Iteration (Frequency) at which we perform rollout operation (testing with the environment). | 10 |
| Learning rate | Learning rate for the optimizer. | $2.5 \times 10^{-6}$ |
| Optimizer | Optimizer for training the neural network. | RMSProp |
| Iteration training data transition size | For each iteration, we generate this number of transitions and use it as training data. | 10k |
| Training step frequency | For each iteration, we perform (iteration training data transition size / training step frequency) number of gradient updates. | 4 |
| Mini-batch size | Size of the mini batch data used to train the Q-network. | 32 |
| $\varepsilon_{\text{train}}$ | $\varepsilon$-greedy policy for exploration during training. | 0.01 |
| $\varepsilon_{\text{eval}}$ | $\varepsilon$-greedy policy for evaluating Q-regressors. | 0.001 |
| Training calibration parameter | Calibration to adjust the difference between the nodes from the candidate pool $m$ which didn't selected during both the expansion nor the dive phases. The calibration is performed based on the average difference between the frontier nodes and their parents. We denote this difference as $\triangle$. | $2.5\triangle$ |
| Discount factor $\gamma$ | Discount factor during the training process. | 0.99 |

Table 6: Hyperparameters for CONQUR training and evaluation.

| Hyperparameters | Description | Value |
|---|---|---|
| Splitting factor $c$ | Controls the number of children created from a parent node | 2 |
| Candidate pool size $m$ | Pool of candidate leaf nodes for selection into the dive or expansion phase | 38 |
| Maximum frontier nodes $F$ | Maximum number of child leaf nodes for the dive phase | 8 |
| Top nodes to expand $l$ | Select the top $l$ nodes from the candidate pool for the expansion phase. | 2 |

Table 7: Different hyperparameters for CONQUR (8 nodes) training and evaluation.

