# OpenReview forum: "ConQUR: Mitigating Delusional Bias in Deep Q-Learning"
_ICLR.cc/2020/Conference — Reject_

### Official Review · AnonReviewer3 · 2019-10-17
**Official Blind Review #3**

**Rating:** 3

**Review:**

A recent paper by Lu et al introduced delusional bias in Q-learning, an error due to the max in the Bellman backup not being consistent with the policy representation implied by the greedy operator applied to the approximated value function. That work proposed a consistent algorithm for small and finite state spaces, which essentially enumerates over realizable policies. This paper proposes an algorithm for overcoming delusional bias in large state spaces. The idea is to add to the Q-learning objective a smooth penalty term that induces approximate consistency, and search over possible Q-function approximators. Several heuristic methods are proposed for this search, and results are demonstrated in Atari domains.

I found the topic of the paper very interesting - delusional bias is an intriguing aspect of Q learning, and the approach of Lu et al is severely limited to discrete and small state spaces. Thus, tackling the large state space problem is worthy and definitely not trivial.

The authors’ proposed solution of combining a smooth penalty for approximate consistency and search over regressors makes sense. The implementation of the search (Sec 3.4) is not trivial, and builds on a number of heuristics, but given the difficulty of the problem, I expect that the first proposed solution will not be straightforward.

I am, however, concerned with the evaluation of the method and its practicality, as reflected by the following issues:
1. The method has many hyper parameters. The most salient one, \lambda, the penalty coefficient, is changed between 0.25 to 2 on the consistency penalty experiment, and between 1 to 1000 in the full ConQUR experiments. I did not understand the order of magnitude change between the experiments, and more importantly, how can one know a reasonable \lambda, and an annealing schedule for it in advance.
2. I do not understand the statistical significance of the results. For example, with the constant \lambda=0.5, the authors report beating the baseline in 11 out of 19 games. That’s probably not statistically significant enough to claim improvement. Also, only one run is performed for each game; adding more runs might make the results clearer.
3. The claim that with the best \lambda for each game, the method outperforms the baseline in 16 out 19 games seems more significant, but testing an optimal hyper parameter for each game is not fair. Statistically speaking, *even if the parameter \lambda was set to a constant zero* for the 5 runs that the method is tested on, and the best performing run was taken for evaluation against the baseline, that would have given a strong advantage to the proposed method over the baseline….
4. For the full ConQUR, there are many more hyper parameters, which I did not understand the intuition how to choose. Again, I do not understand how the results establish any statistically significant claim. For example, what does: “CONQUR wins by at least a 10% margin in 20 games, while 22 games see improvements of 1–10% and 8 games show little effect (plus/minus 1%) and 7 games show a decline of greater than 1% (most are 1–6% with the exception of Centipede at -12% and IceHockey at -86%)” mean? How can I understand from this that ConQUR is really better? Establishing a clearer evaluation metric, and using well-accepted statistical tests would greatly help the paper. At the minimum, add error bars to the figures!
5. While evaluating on Atari shows applicability to large state spaces, it is hard to understand from it whether the (claimed) advantage of the method is due to the delusional bias effect, or some other factor (like implicit regularization due to the penalty term in the loss). In addition, it is hard to understand the different approximations in the method. For example, how does the proposed consistency penalty approximate the true consistency? These could all be evaluated on the simple MDP example of Lu et al. I strongly advise the authors to add such an evaluation, which is easy to implement, and will show exactly how the approximations in the approach deal with delusional bias. It will also be easier to demonstrate the effects of the different hyper parameters in a toy domain.

**Experience Assessment:**

I have published in this field for several years.

**Review Assessment: Checking Correctness Of Derivations And Theory:**

I did not assess the derivations or theory.

**Review Assessment: Checking Correctness Of Experiments:**

I assessed the sensibility of the experiments.

**Review Assessment: Thoroughness In Paper Reading:**

I read the paper at least twice and used my best judgement in assessing the paper.

---

> ### Author Response · Authors · 2019-11-09
> **Response to Review #3 (Part 2 of 2)**
>
> [STATISTICAL SIGNIFICANCE TESTING] While it is not standard in the Atari RL literature to perform statistical significance tests, we ran a Welch’s t-test on performance from iteration 40 to 100, and obtained the following results:
>
> With lambda=10: out of 59 games, 40 games give statistically significant difference (p-val < 0.05) between ConQUR and the baseline. In 31 games, ConQUR is significantly better, while in 9 games the baseline is.
>
> With lambda=1: 31 give statistically significant difference (p-val < 0.05) between ConQUR and the baseline. In 28 games, ConQUR is significantly better, while in 3 games the baseline is.
>
> We also ran a one-sample t-test to compare against pre-trained DQN:
>
> With lambda=10: out of 59 games, 51 games give statistically significant difference (p-val < 0.05) between ConQUR and the pre-trained DQN. In 44 games, ConQUR is significantly better, while in 7 games the pre-trained DQN is.
>
> With lambda=1: 53 give statistically significant difference (p-val < 0.05) between ConQUR and the pre-trained DQN. In 43 games, ConQUR is significantly better, while in 10 games the pre-trained DQN is.
>
>
> 5. Thanks for raising this point about whether our methods are providing improvements because of delusion mitigation or for other reasons. We have provided a detailed response to review 2 that addresses this point: the attribution of improved performance is effectively “by definition” due to the (partial) removal of delusional bias. We acknowledge that this point should have been made more explicitly in the paper and we will clarify in revision.
>
> Regarding “approximations” to consistency: we assume this refers to the soft-consistency penalty only (there are no other approximations other than limiting the search to a subset of possible action assignments). In the paper we explain that the soft consistency penalty measures the degree to which consistency constraints are satisfied: full consistency incurs no penalty while the penalty increases linearly in the degree of violation (other penalty functions are possible of course). Your question also suggests that understanding how much more or less stringent consistency enforcement impacts induced policy quality is important---we agree fully. Our experiments with different values of lambda get partly at this. Evaluating soft consistency vs. *exact* consistency is more challenging in larger domains like Atari due to runtime bottlenecks (large linear programs for linear approximators and solving NP-hard classification problems for DNNs). But we can do so on smaller toy domains (see next paragraph).
>
> We do have results on the simple MDP of Lu et al. with a simplified ConQUR algorithm (with exact consistency checking) and can include these in the paper (we will do so in an appendix). Your suggestion to test how (different degrees of) soft consistency impact the final result vis-a-vis exact consistency is a very nice one, and we will test and explicate this on the simple MDP of Lu et al, or another small example.

---

> ### Author Response · Authors · 2019-11-09
> **Response to Review #3 (Part 1 of 2)**
>
> Thank you for the constructive feedback and for the detailed questions regarding our experiments. Some brief responses to each of your numbered points in turn.
>
> 1. [WHY ORDER OF MAGNITUDE CHANGE] The key difference between (1) the consistency-penalty experiment and (2) the full ConQUR experiments is that the former maintains a single Q-regressor, while the latter maintains multiple Q-regressors. Thus in setting (1), if one makes strong policy commitments early in training, they cannot be undone (there is no search or “backtracking”). In such a case, we want to be less stringent in enforcing policy commitments. In setting (2), we can be more aggressive in enforcing policy commitments, since if they induce poor performance, alternative hypotheses are in play. (In principle, with “exhaustive” search, per Lu et al. 2018, this will find the optimal policy-consistent value function.) Nevertheless, Fig. 11 on p. 20 (Appendix D.3) shows lambda=1, 10 performs the best (or comparably). Larger lambdas are not necessary.
>
> [SELECTING LAMBDA] Selecting a reasonable fixed lambda is similar to selecting regularization parameters in supervised learning—-cross-validation or other approaches may be used.
>
> Annealing: lambda is gradually increased from 0 to the final value since we do not wish to over-constrain the Q-regressor with potentially bad policy commitments near the start of training. We chose a simple schedule: lambda = final_value * step / (step + 200k), which reaches half of the final value at step 200k. We will elaborate on this in the paper. Other ways of tuning are of course possible.
>
> 2. We agree with this point and will make revisions accordingly. We will update our figures in the revised paper to include the mean score over reruns (most games are re-run with 3-5 random trials) and error bars of the 95% confidence interval. (An updated version of the main figures can be seen here: https://tinyurl.com/ryzyhrr ). Our conclusions about ConQUR are not impacted by this:
>
> DQN(lambda = 0.5): 10 wins, 3 losses, 6 inconclusive (see dqn_reg_0_5.pdf)
> DDQN(lambda = 0.5): 9 wins, 2 losses, 8 inconclusive (see ddqn_reg_0_5.pdf)
>
> This suggests using a single soft-penalty constant generally does not hurt performance and can improve over baseline in a non-trivial fraction of the Atari environments.
>
> 3. As discussed in Question 2 above, lambda=0.5 works well across all tested games even without taking the max over runs). In general, Q-learning (with or without a consistency penalty) behaves differently in different environments, thus games will have their own optimal penalty constants. Practitioners often select good hyperparameters for their particular task/game (see above comment on selecting hyperparameters), and this is often seen in the literature. That said, we will make clearer the role/value of using just a single, fixed lambda.
>
> 4. As above, we agree with this general point on statistical significance (we discuss details below on how we address this).
>
> [HYPERPARAMETER TUNING] The full set of ConQUR hyperparameters (fixed across all games) is shown in Table 6, Appendix D. There are 7 additional hyperparameters beyond those used in standard DQN hyperparameters (e.g. eps_train, discount factor, etc., which we match to standard values used in DQN implementations in the Atari RL literature). Five of the seven hyperparameters relate to easy-to-understand search tree parameters, including branching factor, etc. The two remaining hyperparameters (Boltzmann iteration, training calibration parameter for scoring function) will need additional insight and potential tuning (e.g., see our discussion in response to review 2 regarding parameter lambda), but should not be more cumbersome to tune than other deep learning architectures. Due to GPU resource limitations, we did not explore the full range of hyperparameter combinations.
>
> [DESCRIPTION OF RESULTS] First, one brief clarification: the statements such as “ConQUR wins by a 10% margin…” are not intended to be statistical claims, rather they are descriptions of the larger table of results from the appendix. Per the point raised by reviewer 1, we will attempt to get the full table of results into the main text so some of this descriptive text can be condensed.
>
> [EVALUATION METRICS] Our evaluation metric (game score) is the standard for Atari RL benchmarks. We agree with your point about statistical validity of our conclusions. We have results evaluated using 5 random seeds per game, and will include the results in the revised paper (3 to 5 random seeds is standard across Atari RL papers.) The 5-seed results, showing mean and 95% confidence intervals can be seen at: https://tinyurl.com/yz5xy5ox . We were unable to run on multiple seeds prior to submission due to limited GPU resources, our apologies for that.

---

### Official Review · AnonReviewer2 · 2019-10-21
**Official Blind Review #2**

**Rating:** 3

**Review:**

This paper presents a solution to tackling the problem of delusional bias in Deep Q-learning, building upon Lu et.al. (NeuRIPS 2018).  Delusional bias arises because independently choosing maximizing actions at a state may be inconsistent as the backed-up values may not be realizable by any policy. They encourage non-delusional Q-functions by adding a penalty term that enforces that the max_a in Q-learning chooses actions that do not give rise to actions outside the realizable policy class. Further, in order to keep track of all consistent assignments, they pose a search problem and propose heuristics to approximately perform this search. The heuristics are based on sampling using exponentiated Q-values and scoring possible children using scores like Bellman error, and returns of the greedy policy. Their final algorithm is evaluated on a DQN and DDQN, where they observe some improvement from both components (consistency penalty and approximate search).

I would lean towards being slightly negative towards accepting this paper. However, I am not sure if the paper provides enough evidence that delusional bias is a very relevant problem with DQNs, when using high-capacity neural net approximators. Further, would the problem go away, if we perform policy iteration, in the sense of performing policy iteration instead of max Q-learning (atleast in practice)? Maybe, the paper benefits with some evidence answering this question. To summarize, I am mainly concerned about the marginal benefit at the cost of added complexity and computation for this paper. I would appreciate more evidence justifying the significance of this problem in practice.

Another comment about experiments is that the paper uses pre-trained DQN for the ConQur results, where only the last linear layer of the Q-network is trained with ConQur. I think this setting might hide some properties which arise through the learning process without initial pre-training, which might be more interesting. Also, how would other auxilliary losses compare in practice, for example, losses explored in the Reinforcement Learning with Auxilliary Tasks (Jaderberg et.al.) paper?

**Experience Assessment:**

I have read many papers in this area.

**Review Assessment: Checking Correctness Of Derivations And Theory:**

I did not assess the derivations or theory.

**Review Assessment: Checking Correctness Of Experiments:**

I assessed the sensibility of the experiments.

**Review Assessment: Thoroughness In Paper Reading:**

I read the paper at least twice and used my best judgement in assessing the paper.

---

> ### Author Response · Authors · 2019-11-09
> **Response to Review #2**
>
> Thank you for the constructive feedback and for raising some important questions. Some brief responses to specific points/questions you raise.
>
> [DOES DELUSION ARISE IN PRACTICE?] The purpose of the experiments is to show that mitigating delusional bias, even with high-capacity NNs, can offer improvements. We believe the experiments show that delusional bias does occur in practice since the pre-trained Q-regressors upon which we improve are Dopamine-trained DQNs/DDQNs. Our methods differ only from (say) DQN in the use of the soft-consistency penalty (plus the use of search to explore multiple assignments against which to apply this penalty). We claim that this tackles only “policy inconsistency” (i.e., delusion). Because we obtain improvements over the pre-trained DQNs, our conclusion is that delusion does, indeed, arise in practice. In retrospect, we should have made this important point much more explicit in the paper---our apologies for not doing so originally---and we will do so in revision.
>
> Our experiments, we believe, do not demonstrate the full power of removing delusion, since we only retrain the last FC layer of the pre-trained DQN, which in fact limits our performance opportunities vs. training on all layers---this alone results in significantly better greedy policies in many instances.
>
> [WILL DELUSION ARISE IN POLICY ITERATION?] This is a good point, and while it may depend on the implementation, generally policy iteration will not have delusional bias. However, our contribution is focused on improving “pure” value-based methods like Q-learning (and related methods like DDQN). These are widely used algorithms, that researchers and practitioners often have strong reasons to use---our focus is to mitigate delusional bias to extract maximum value from such methods whenever they are used.
>
> [WHY USE PRE-TRAINED NETWORKS]? The rationale for improving with pre-trained DQNs is three-fold.
>
> First, it demonstrates that delusion actually causes problems in practice (as discussed above, we will articulate this point much more explicitly in revision). In some sense, by freezing the feature representation learned by DQN, and demonstrating that a “linear” value function over those same features can be trained in (partially) non-delusional  fashion to extract improvements gives more of a focus on non-delusional training (as opposed to novel “feature discovery”).
>
> The second reason is a practical one---it allowed us to scale our experiments to cover a range of hyperparameters and run the entire Atari suite (rather than selecting just a few high-performing games).  We completely agree with your broader point about experimenting with our methods with full network training (i.e., from scratch) to understand their performance. In some sense, this paper provides a (we hope, compelling) first exploration of these ideas.
>
> Third, from a practical point of view, this “linear tuning” approach offers a relatively inexpensive alternative to extract improvements from a model learned using classic techniques (e.g., linear tuning requires many fewer training samples).
>
> We also note that, if the full application of ConQUR is too expensive in some settings, adding a our simple consistency penalty can sometimes provide a lift (and rarely hurts), as shown by the experiments in Section 4.1. This requires no major changes to standard DQN, DDQN or the like and adds no significant implementation complexity or computational cost.
>
> [AUXILIARY LOSSES] Thanks for making the reference to auxiliary losses, this is an interesting question. While our penalty term focuses on consistency for a single task and auxiliary losses help accelerate learning for the main task, we can imagine applying a consistency penalty for each auxiliary objective (in addition to minimizing task’s Bellman error). This direction is interesting to explore, which we will cite in the revised paper.

---

### Official Review · AnonReviewer1 · 2019-10-28
**Official Blind Review #1**

**Rating:** 6

**Review:**

This paper focuses on addressing the delusional bias problem in deep Q-learning, and propose a general framework (ConQUR) for integrating policy-consistent backups with regression-based function approximation for Q-learning and for managing the search through the space of possible regressors. Specifically, it proposes a soft consistency penalty to alleviate the delusional bias problem while avoiding the expensive exact consistency testing. This penalty encourages the parameters of the model to satisfy the consistency condition when solving the MSBE problem.

Pros:
The soft penalization itself is already shown to be effective in further improving any regression-based Q-learning algorithms (e.g., DQN and DDQN). When combining the soft consistency penalty and the search procedure, it is shown to significantly outperform the DQN and DDQN baselines, which is impressive. This work presents novel idea and solid supporting experiments.

Cons:
The major experimental results of the paper are in Table 4 of Appendix D. This is not a good effort to save space by moving the most important results into appendix. At least a selected set of results should be presented in the main paper rather than in appendices. One alternative approach is the authors plot a bar figure to demonstrate the performance of different algorithms on different Atari games.


Other comments:
•	Fig. 2 two is a bit hard to read due to too many curves.
•	Standard baseline results other than DQN and DDQN should also be listed, in order to demonstrate that solving delusional bias could make Q-learning more competitive than alternatives (e.g., A3C, PPO, TRPO).


**Experience Assessment:**

I have published in this field for several years.

**Review Assessment: Checking Correctness Of Derivations And Theory:**

I assessed the sensibility of the derivations and theory.

**Review Assessment: Checking Correctness Of Experiments:**

I carefully checked the experiments.

**Review Assessment: Thoroughness In Paper Reading:**

I read the paper thoroughly.

---

> ### Author Response · Authors · 2019-11-09
> **Response to Review #1**
>
> Thank you for the constructive feedback. Some brief responses:
>
> We will try to fit in the results of Table 4 into the main text---if this is not feasible due to space constraints, we’ll include a comprehensive summary as you suggest.
>
> With respect to other algorithms (A3C, PPO, TRPO, etc.) we will add a brief comparison to such algorithms in the literature, thanks for the suggestion. While we don’t make a broad claim that removing delusion will make Q-learning competitive universally---we believe some domains may be better suited to policy-based or actor-critic-style algorithms---it does in a number of the cases examined here. However, there are many instances where Q-learning is desirable for other reasons, and our primary aim is to try to extract maximum performance from Q-learning itself. We will make this part of our motivation more explicit.
>
> We will make Fig. 2 more readable---apologies for that!

---

### Decision · Program_Chairs · 2019-12-19

**Decision:**

Reject

**Comment:**

While there was some support for the ideas presented, the majority of reviewers felt that this submission is not ready for publication at ICLR in its present form.

Concerns raised included the need for better motivation of the practicality of the approach, versus its computational cost. The need for improved evaluations was also raised.